# Optimization of Infrared Heating Conditions for Precooked Cowpea Production Using Response Surface Methodology

**DOI:** 10.3390/molecules26206137

**Published:** 2021-10-11

**Authors:** Opeolu M. Ogundele, Sefater Gbashi, Samson A. Oyeyinka, Eugenie Kayitesi, Oluwafemi A. Adebo

**Affiliations:** 1Department of Biotechnology and Food Technology, Faculty of Science, Doornfontein Campus, University of Johannesburg, Johannesburg P.O. Box 17011, South Africa; sefatergbashi@gmail.com (S.G.); sartf2001@yahoo.com (S.A.O.); 2Department of Consumer and Food Science, University of Pretoria, Private Bag X20, Pretoria 0028, South Africa

**Keywords:** cowpea, infrared heating, optimization, preconditioning, pre-cooked, RSM

## Abstract

The infrared heating of preconditioned cowpea improves its utilization and potential application in food systems. This study investigated the effect of optimizing preconditioning and infrared heating parameters of temperature and time on cooking characteristics of precooked cowpeas using response surface methodology (RSM). The moisture level (32–57%), infrared heating temperature (114–185 °C), and time of processing the seeds (2–18 min) were optimized using a randomized central composite design to achieve optimal characteristics for bulk density and water absorption. A second-order polynomial regression model was fitted to the obtained data, and the fitted model was used to compute the multi-response optimum processing conditions, which were the moisture of 45%, the heating temperature of 185 °C, and time of 5 min. Precooked cowpea seeds from optimized conditions had a 19% increase in pectin solubility. The total phenolic and total flavonoid contents were significantly reduced through complexation of the seeds’ phenolic compounds with other macromolecules but nonetheless exhibited antioxidant properties capable of scavenging free radicals. There was also a significant reduction in phytate and oxalates by 24% and 42%, respectively, which was due to the heat causing the inactivation of these antinutrients. The obtained optimized conditions are adequate in the production of precooked cowpea seeds with improved quality.

## 1. Introduction

Cowpea (*Vigna unguiculata* L. Walp) is a underutilized crop indigenous to Africa, with an outstanding potential to contribute to the major challenges in food and nutrition security as well as in agricultural sustainability [1]. Cowpea has good nutritional value in terms of a high protein content of about 18–30% [2,3], minerals, such as iron, zinc, calcium, and magnesium [4] and health-promoting constituents, particularly phenolic compounds, such as flavonoids and phenolic acids, which are known for their antioxidants and free radical scavenging activities [5]. However, the presence of antinutrients, such as trypsin inhibitors and phytic acids, is important in the bioavailability of micronutrients, such as iron and zinc, limiting their use, as it does with other legumes [6]. Cowpea is mostly utilized as cooked whole seeds often achieved after boiling for up to 2 h, resulting in extensive energy consumption and long food preparation times [7]. Owing to this, attempts are being made to reduce the time it takes to cook leguminous seeds by employing traditional methods, such as soaking in salted water using Kawe (natural rock salt) and the application of hydrothermal process including microwaving annealing and infrared heating [8].

Infrared radiation heating is a heat pre-treatment process that uses electromagnetic radiation in the infrared region within the wavelength of about 3 to 1000 µm [8,9]. Over the last decade, attempts have been undertaken to rapidly heat moisture-conditioned legumes, using a table-top infrared heater with varied processing parameters of moisture (30–41%), infrared heating temperature (130–170 °C), and time (3–8 min) of bambara groundnut [10] and cowpea [11] seeds, with results suggesting an improvement in water hydration during soaking and cooking as well as a reduction in cooking time. Other cooking characteristics of infrared heated cowpea grains include sensory attributes, such as color, taste, and texture. Cooking time is an essential characteristic commonly used to determine the quality of a good legume [12]. However, the increased water absorption by infrared heated cowpeas during cooking has also been associated with reducing the time of cooking but increasing the incidence of splits [13]. In addition, the lower bulk density in infrared heated cowpea grains implied more air space in the cotyledon, which enhances water absorption during cooking, and the disintegration of the middle lamella, as evidenced by pectin solubility and the separation of cells along the cell wall during cooking of cowpeas, is one of the major physicochemical and structural changes that must occur prior to the softening of cowpeas [14]. Similarly, research has been done to explore how infrared heat treatments affect the nutritional, antinutritional, and health-promoting aspects of legume bean seeds [7], addressing hard to cook defects [14] and the development of ready-to-eat African porridge for feeding young children [15].

Notwithstanding, several studies have pointed out the drawbacks of utilizing manual table-top infrared heaters, specifically their inability to determine the relative influence of heat loss on the structure and molecular components of legume seeds during infrared heating processing [13,16]. To address this challenge, optimization of the processing conditions of infrared heating (using a table-top infrared heater) to enhance cowpea seeds’ quality was investigated in this study. According to Adebo, Njobeh [17], response surface methodology (RSM) is a widely established optimization strategy that uses a mixture of statistical and mathematical approaches to find the best circumstances (values) of variables for desired responses (parameters). This procedure has been useful in optimizing conditions of various food processes including soaking, fermentation, and thermal processes to enhance food quality and safety [17,18]. As a result of its multiple advantages, the central composite design (CCD), among other RSM approaches, is one of the most often utilized numerical optimization techniques in food processing [19]. To the best of our knowledge to date, the amount of information available about optimizing the moisture preconditioning conditions prior to infrared treatment, as well as the subsequent infrared heating temperature and time, is still limited, but it is critical for improving the cooking characteristics as well as the nutritional and health-promoting qualities of legumes as an important source of protein. Therefore, this study aimed to investigate the effect of optimizing the processing parameters of infrared heaters on some of the cooking characteristics, antinutritional properties, phenolic contents, as well as the antioxidants activities of cowpeas. In this paper, the preconditioning and infrared heating parameters (temperature and time) of cowpea seeds were optimized using a randomized central composite design (CCD), and infrared heated cowpea seeds with improved cooking and nutritional quality were developed.

## 2. Materials and Methods

### 2.1. Materials

Cowpea grains grown in South Africa (variety: Agrinawa) were obtained from the Agricultural Research Council, Institute for Tropical and Subtropical Crops, Nelspruit, South Africa. The cowpea seeds were manually sorted to remove defective seeds and stored at 4 °C until when needed for moisture preconditioning and used for infrared heat treatment. All chemicals and solvents used were laboratory grade.

### 2.2. Experimental Design

A set of experiments was statistically designed based on RSM-CCD using Statistica version 7 statistical software (StatSoft, Tulsa, OK, USA). The independent factors of pre-treatments studied were level of moisture (X_1_), infrared heating temperature (X_2_), and time (X_3_), with intervals of 32–57%, 114–185 °C, and 2–18 min, respectively. Fifteen (15) experimental runs were generated from the combination of factors (Table 1). The mathematical model represented in Equation (1) [17,19] describes the relationship between the variables [20].
(1)$$Y=\beta_{0}+\beta_{1}A+\beta_{2}B+\beta_{3}C+\beta_{12}AB+\beta_{13}AC+\beta_{23}BC+\beta_{12}ABC+\beta_{11}A^{2}+\beta_{22}B^{2}+\beta_{33}C^{2}$$
where *Y* is the response, $\beta_{0},\;\beta_{1},\;\beta_{2},\;\ldots \beta_{33}$ are the coefficients of regression, while *A*, *B*, and *C* are the independent experimental factors. The 3D surface plots and the model-fit factor effects were used to show the effects of processing parameters on the responses with respect to the preset optimization conditions. Adequacy of the model fit was determined by evaluating various model-fit validation parameters, coefficient of determination (*R*^2^), and adjusted coefficient of determination (*R*^2^*adj*) [17,19]. Model significance was determined at a probability level of 95% (*p* < 0.05). The various mathematical functions used to compute these parameters are presented in Equations (2) and (3) [17,19].
(2)$$R^{2}=\frac{\sum_{i=1}^{n}{\left({\hat{y}}_{i}-\bar{y}\right)}^{2}}{\sum_{i=1}^{n}{\left(y_{i}-\bar{y}\right)}^{2}}$$
(3)$$R^{2}adj=1-\frac{k-1}{k-p}\left(1-R^{2}\right)$$
where *R*^2^ is the coefficient of determination, *n* is the sample size, *ȳ* is the estimated mean value, *y_i_* is the experimental value, *ŷ* is the predicted value, *R*^2^*adj* is the adjusted coefficient of determination, *p* is the number of regression coefficients, and *k* is the total number of observations.

After the model fit and validation of the model adequacy, the resultant quadratic optimization models were used for optimization and computation of the global optimal conditions for the pre-processing of cowpea samples using the global optimization feature in the Minitab 17 statistical software (State College, PA, USA). Optimum conditions were obtained by minimization (bulk density) and maximization (water absorption capacity) of responses to achieve the highest desirability factor of 1.

### 2.3. Moisture Conditioning and Infrared Heating Pre-Treatment of Cowpea Seeds

The moisture preconditioning was done at levels of 32, 40, 45, 54, and 57% (dry basis) prior to infrared heat treatment, which was carried out as described by Mwangwela and Waniska [13]. Cowpeas (500 g) at an initial moisture content of 11%, were conditioned to 156, 243, 310, 469, and 537 g/kg moisture content (wet basis) by steeping in deionized water, shaking for 6 h, and holding for 12 h at 22 °C for the moisture to equilibrate throughout the seeds to moisture levels of 32, 40, 45, 54, and 57%, respectively. A closed system infrared heater (MW184, Delphius Technologies, Pretoria, South Africa) was used for the current work. The infrared heater was first pre-heated for 20 min. Then, the preconditioned whole cowpea samples were heated between 114 and 185 °C for 2–18 min in a closed infrared heater for each experimental run (Table 2), after which the seeds were dried in a hot air dryer (D-37530, Thermo Fischer Scientific, Pretoria, South Africa) at 50 °C for 12 h. Then, the samples were cooled to room temperature (22 ± 2 °C) for 1 h. Part of the infrared heated seeds were milled to pass through a 500 µm aperture sieve using a Retsch ultra centrifugal mill (Model ZM200, Haan, Germany). The infrared heated cowpea seeds and flours were stored in an airtight container and kept at 4 °C until further analyses. The control sample was flour from preconditioned whole cowpea, which was then dried in a hot air dryer.

### 2.4. Physicochemical and Functional Analyses

#### 2.4.1. Bulk Density

The bulk density of seed samples at different conditions was determined as described by Alves and Da Silva [21]. Grain samples (50 g) were weighed into a 100 mL graduated measuring cylinder. On a laboratory bench, the cylinder was gently tapped numerous times to achieve a steady volume, which was then recorded. The bulk density (g/mL) was estimated as the weight of the sample per unit volume of the sample after tapping.

#### 2.4.2. Water Absorption Capacity

The water absorption capacity of seed samples at different conditions during soaking were determined as described by [10]. First, 10 g of seeds were put in 500 mL Erlenmeyer flasks with 60 mL deionized water in each flask. The flasks were put in an incubator for 24 h at 25 °C. After soaking, surplus water was passed through a metal sieve with a 2.5 mm aperture, and the seeds were wiped dry and weighed. The gain in weight was expressed as g water absorbed per kg cowpea seeds.

#### 2.4.3. Soluble Pectin

Soluble pectin was determined in cowpea seed samples using the method described by Ndungu and Emmambux [14] with modification. Milled flours (5 g) were first treated with ethanol (95% *v*/*v*) to remove soluble sugar. The mixture was mixed for 10 min before being centrifuged (5702 R-Eppendorf Centrifuge, Hamburg, Germany) for 10 min at 17,300× *g*. The remaining pellet was extracted twice with 40 mL of 95% *v*/*v* ethanol, which was followed by a final extraction with 40 mL of absolute ethanol. Following that, the pellet (alcohol-insoluble particles) was vacuum dried at room temperature (25 °C) and stored in a desiccator. The alcohol-insoluble solids (AIS) sample was extracted with hot water, and the extracts were considered as hot water-soluble pectin (HWSP). The determination of hot water-soluble pectin (HWSP) was conducted as described by [14] using 1 g of the AIS. The pectin content of each fraction was expressed as galacturonic acid as determined by the meta-hydroxydiphenyl method of Blumenkrantz and Asboe-Hansen [22] at 95 degrees Celsius using galacturonic acid as a standard.

### 2.5. Total Phenolic, Total Flavonoid Content, and Antioxidant Capacity Assays

#### 2.5.1. Extraction Procedure

Extracts from the samples for total phenolic contents, total flavonoid contents, and antioxidant capacity were prepared using acidified methanol (1% HCl in 80% *v*/*v* methanol) as described by Kayitesi [7]. Each sample (3 g) was extracted with 30 mL solvent in three phases as follows: first, 10 mL solvent was added to 3 g of the sample in a conical flask, stirred for 3 h, and centrifuged (5702 R-Eppendorf Centrifuge, Hamburg, Germany) at 3500 rpm for 10 min at ambient temperature (25 °C), and the supernatant was decanted. The sample residue was rinsed with 10 mL of the solvent and agitated for 20 min before centrifuging and decanting the supernatant as before. This procedure was followed again. The supernatants were preserved in an airtight glass bottle with aluminum foil and maintained at 4 °C until they were analyzed.

#### 2.5.2. UHPLC Identification and Quantification of Phenolic Compounds

The identification and quantification of phenolic compounds were achieved using an ultra-high performance liquid chromatography system coupled to a photodiode array detector (UHPLC-PDA). The instrument (Shimadzu Kyoto, Japan) consisted of an auto-sampler (SIL-40C_XR_), degassing unit (DGU-403), solvent delivery module (LC-40B_XR_), and column oven (CTO-40C) interfaced with a PDA detector (SPD-M40). A Waters Symmetry C_18_ column of dimensions 5 µm, 4.6 × 250 mm (Waters, Milford, MA, USA) was used for the separation of the analytes at a constant mobile flow rate of 0.5 mL/min and temperature of 40 °C. The mobile phases were A: 0.1% formic acid in the water, and B: 0.1% formic acid in methanol/acetonitrile (50:50, *v*/*v*). The gradient elution program started with 10% solvent B for 0.5 min, gradually ramped up to 95% solvent B within 7 min, increased to 100% B within 1 min, was kept constant at this condition (i.e., 100% B) for 1 min, decreased to 90% B within 0.5 min, further dropped to 10% B in 1 min, and was left to equilibrate at this condition (i.e., 10% B) for 4 min. The total sample run time was 15 min. The PDA detector was set to acquire over the wavelength range of 220 to 500 nm at a sampling frequency of 1.5625 Hz. Analytes were identified using the UV wavelengths and retention times of the pure compounds, which were as follows: kaempferol (366 nm and 11.134 min), luteolin (348 nm and 10.631 min), ferulic acid (322 nm and 9.444 min), taxifolin (290 nm and 9.289 min), apigenin (336 nm and 11.327 min), quercetin (371 nm and 10.599 min), *p*-coumaric acid (309 nm and 9.378 min), sinapic acid (319 nm and 9.026 min), caffeic acid (323 nm and 8.596 min), gallic acid (271 nm and 8.533 min), and vanillic acid (261 nm and 8.707 min). Linear calibration curves were generated for each of the target analytes over a concentration range 0.5–40 µg/mL and used for the quantification of the analytes.

#### 2.5.3. Total Phenolic Content (TPC)

The total phenolic content of the extracts was determined spectrophotometrically using the Folin–Ciocalteu procedure modified for a 96-well plate, as described by Ainsworth and Gillespie [23]. Ten microliters (10 µL) of gallic acid standard or an aliquot of the extract (10 µL) was pipetted into microplate wells, which was followed by 50 µL of Folin–Ciocalteu reagent and 50 µL of sodium bicarbonate (Na_2_CO_3_). The microplate was wrapped in foil and kept at 25 °C for 30 min in the dark. The absorbance of standard and extracts was determined at 750 nm with an iMark microplate absorbance reader (Bio-Rad laboratories 168-1130, Hercules, CA, USA) and expressed as mg catechin equivalent/g.

#### 2.5.4. Total Flavonoid Content (TFC)

The determination of the total flavonoid content of the extracts was done by using aluminum chloride assay, as suggested by Kalita and Tapan [24]. First, 10 µL of the standard and the extracts were pipetted into microplate wells; then, 30 µL of NaNO_2_ was added, which was followed by 30 µL of AlCl_3_ and 100 µL of NaOH. Furthermore, using an iMark microplate absorbance reader (Bio-Rad laboratories 168-1130, Hercules, CA, USA), the absorbances of the extracts and quercetin were determined at 450 nm and were expressed as mg catechin equivalent/g.

#### 2.5.5. Ferric Reducing Antioxidant Power (FRAP) Assay

The free radical scavenging assay was determined as described by Adedapo and Jimoh [25]. FRAP reagent in 25 mL of acetate buffer (300 mM, pH 3.6), 2.5 mL of tripyridyltriazine (10 mM), and 2.5 mL of FeCl_3_·6H_2_O (20 mM) was warmed and kept in a water bath at 37 °C. Then, 30 μL volume of 1 mg/mL sample was mixed with 900 μL FRAP solution, and then, the mixtures were allowed to stand in the dark for 30 min. Using a microplate absorbance reader (iMark 168–1130, Bio-Rad laboratories, Hercules, CA, USA), the absorbance of the colored result, a ferrous tripyridyltriazine complex, was measured at 595 nm and was expressed as mg gallic acid equivalent/g.

### 2.6. Antinutritional Factors (ANFs) Analyses

#### 2.6.1. Total Phytic Acid Content

Phytic acid content was determined as described by Gao and Shang [26]. The phytic acid content of flour from seed samples was determined using the Wade reagent prepared by the addition of 0.03% FeCl_3_·6H_2_O and 0.3% sulfosalicyclic acid (C_7_H_6_O_6_S). Then, 2.4% HCl solution was mixed with 0.3 g samples and was shaken in an orbital shaker (Labotec 261, Midrand, South Africa) at 220 rpm for 6 h. After that, 0.1 mL of clear supernatant was combined with 3 mL of water and 0.2 mL of Wade reagent before being centrifuged for 10 min at 5500 rpm. The absorbance was determined using a Spectrophotometer (Boeco S-20, Hamburg, Germany) at 500 nm with distilled water as a blank and represented as mg/g phytic acid/flour sample.

#### 2.6.2. Oxalate Content

Oxalates were determined as described by Muchoki and Lamuka [27]. Briefly, 3 mg of sodium oxalate was dissolved in 10 mL of H_2_SO_4_ (0.5 M) to make a standard sodium oxalate solution. A standard curve was drawn after titration with KMNO_4_ (0.1 M) at 60 °C using a micro-burette produced a faint violet color that was stable for at least 15 s. Then, 0.1 g of a dried sample was extracted with 30 mL of HCl (1 M) in a boiling water bath for 30 min. The sample was cooled and then shaken and filtered using No. 1 Whatman filter paper (diameter, 110 mm). The filtrate was adjusted to a pH of about 8.5 to 9 with ammonium hydroxide (8 M) followed by re-adjusting it to pH 5.0–5.2 with acetic acid (6 N). An aliquot of 10 mL was precipitated with 0.4 mL of 5% CaCl_2_, shaken thoroughly, allowed to settle at 22 °C for 16 h, and centrifuged (5702 R-Eppendorf Centrifuge, Hamburg, Germany) at 3000 rpm for 15 min. The supernatant was discarded, rinsed twice with 2 mL of ammonium hydroxide (0.35 M), and then, the cake (pellet) drip-dried. The pellet was dissolved in 10 mL of H_2_SO_4_ (0.5 M) followed by titration with KMNO_4_ (0.1 M) at 60 °C using a micro-burette to a faint violet color that was stable for at least 15 s. Oxalates content in the sample was determined from the standard curve and were expressed as mg/100 g.

### 2.7. Statistical Analysis

The experimental work was repeated three times, and analyses were done in triplicate. Data were analyzed using one-way analysis of variance (ANOVA), and means were compared using the Fisher Least Significant Difference (LSD) test (*p* ≤ 0.05) using Statistica version 11 (StatSoftInc, Tulsa, OK, USA).

## 3. Results and Discussion

Results for the cooking characteristics of precooked cowpea are presented in Table 2. There are marked differences in the physicochemical characteristics (bulk density and water absorption capacity) of the precooked cowpeas at the various treatment conditions. Residuals of both water WAC and bulk density (Appendix Aa,b) data followed a normal distribution, since the data in the plot of the normal distribution of residuals were close to the line [28].

### 3.1. Modeling and Optimization of the Infrared-Induced Physicochemical Perturbations of Cowpea

In order to describe the empirical relationship between the physicochemical characteristics of the precooked cowpea grains and the infrared heat treatment, the quadratic regression function (Equation (1)) was fitted to the experimental data. The resultant second-order models (Table 3) provide a strong approximation to the true relationship between the response variables (bulk density and water absorption capacity) and the control variables (moisture, temperature, and time). From these models, it was possible to generate the 3D surface plots (Figure 1a–c), which provide a visual representation of the changes in bulk density and water absorption capacity with respect to the control variables. The response surface plot shows that the bulk density decreased with an increase in moisture level, infrared heating temperature, and heating time. On the other hand, an increase in the water absorption capacity (95.45–126.83 g/kg) (Table 2) was recorded with an increase in the parameters (i.e., moisture level, infrared heating temperature, and heating time) (Figure 1d–f).

Previous studies similarly reported a decrease in bulk density and an increase in the water absorption of 40% moisture conditioned cowpea infrared heated at 180 °C for 10 min [13], which was possibly due to the structural and molecular changes in the treated seeds.

### 3.2. Model Validation, Adequacy, and Factor Effects

Following the optimization model fit, the resultant model fitness and adequacy were determined by evaluating the coefficient of determination (*R*^2^). The different models obtained from the model fit representing each response variable are provided in Equations (4) and (5). The *R*^2^ values of the models were above 90 (93.45–99.71%) (Table 3). *R*^2^ values close to 100% as observed in Table 3 indicate that the models can explain/describe a significant percentage (in this case more than 90%) of the patterns in the data. This shows that the models are well fit and adequate. By means of these models, it was also possible to deduce the magnitude and significance of the various factor effects on the response variables (Table 4) [19]. The linear factors of moisture (*x*_1_), temperature $\left(x_{2}\right)$, and time $\left(x_{3}\right)$ for cowpea seeds had a significant (*p* < 0.05) effect on the bulk density of the cowpea samples (Table 4), while only the linear effects of infrared heating temperature $\left(x_{2}\right)$ and time $\left(x_{3}\right)$ had a significant (*p* < 0.05) effect on water absorption capacity. The quadratic effect of all the variables ($x_{1}^{2}$,$\;x_{2}^{2}$, and $x_{3}^{2}$) were significant (*p* < 0.05) except for that of water absorption capacity. For the interactive effects, moisture and time ($x_{1}x_{3}$), as well as infrared heating temperature and time ($x_{2}x_{3}$), had a significant effect on bulk density, whereas only the temperature and time interaction ($x_{2}x_{3}$) had a significant effect (*p* < 0.05) on water absorption capacity.
(4)$$Y_{1}=0.741-0.00833\left(x_{1}\right)+0.02609\left(x_{2}\right)-0.00520\left(x_{3}\right)+0.000064\left(x_{1}^{2}\right)-0.000008\left(x_{2}^{2}\right)+0.000084\left(x_{3}^{2}\right)\;-0.000008\left(x_{1}x_{2}\right)+0.00172\left(x_{1}x_{3}\right)-0.000039\left(x_{2}x_{3\;}\right)$$
(5)$$Y_{2}=1248-18.88\left(x_{1}\right)-9.08\left(x_{2}\right)-4.17\left(x_{3}\right)+0.1699\left(x_{1}^{2}\right)+0.02708\left(x_{2}^{2}\right)+0.3571\left(x_{3}^{2}\right)+0.01952\left(x_{1}x_{2}\right)\;-0.0223\left(x_{1}x_{3}\right)-0.01677\left(x_{2}x_{3}\right)$$

### 3.3. Multi-Response Numerical Optimization

A global (i.e., multi-response) optimization approach was adopted to determine the optimized conditions for infrared processing of the cowpeas. For the optimization, it was desired to minimize the bulk density and maximize the absorption capacity. The multi-response optimization of the process variables was done using Minitab 17 statistical software and the optimum derived conditions were 45% for moisture level, 185 °C for infrared heating temperature, and a time of 5 min. The corresponding predicted parameters at these conditions were bulk density (0.68 g/mL) and water absorption capacity (188.02 g/kg). To confirm the predicted values, the predicted values were tested in triplicate using the optimal infrared conditions obtained. The subsequent analysis gave the following results: bulk density (0.68 g/mL) and water absorption capacity (181.55 g/kg). These obtained results are close to the optimized values obtained, thus showing that the regression models obtained could adequately predict the response parameters.

### 3.4. Pectin Solubility

Pectin solubilization is one of the most important molecular changes during cooking. This process involves the thickening of cell walls and cell separation responsible for the softening and reduction in cooking time of legume seeds [10,13,14]. The optimal conditions investigated in this study significantly increased the pectin solubility by 19%, and the result was higher than the 11% increase in the pectin solubility observed in cowpea pre-treated at a 41% moisture level, 153 °C for 5 min (Figure 2).

The result in this study indicates that a shorter cooking time is achievable when cowpeas are processed at the optimum conditions (45% moisture, 185 °C, and 5 min), as compared to the 14% reduction in cooking time previously reported by other authors on infrared heated cowpeas [13,14]. The findings from our study clearly show that aside from infrared heating improving product quality, this environmentally friendly technology could reduce energy consumption and costs and produce convenient foods with good economic value, as earlier reported [29].

### 3.5. Total Phenolic, Total Flavonoid Content, and Antioxidant Capacity

Cowpeas are a rich source of tannins, flavonoids, phenolic compounds, and antioxidants and thus considered health-promoting foods [30,31]. Table 5 shows the total phenolic content, total flavonoid content, and antioxidant property of extracts of cowpea flour samples. As observed, the infrared heating had a significant (*p* < 0.05) reduction effect on all health-promoting properties but was still capable of scavenging free radicals (Table 5). The total phenolic content, total flavonoid content, and antioxidant activities (FRAP) reduced by approximately 57%, 61%, and 72%, respectively.

It is important to note that the total phenolic content refers to the content of antioxidants in this study because phenolic compounds are not the only ones that react with Folin–Ciocalteu reagent. According to [32], other compounds, such as vitamin C, some amino acids, peptides, reducing sugars, organic acids, Maillard reaction products, and some more compounds could react with Folin–Ciocalteu reagent and therefore contributing to the total phenolic content.

Excessive splitting due to infrared heating and cooking of cowpea was reported to facilitate leaching out of the phenolic components in the samples, which explains why losses in phenolic compounds were generally observed [7,16]. Luthria and Pastor-Corrales [33] reported that phenolic compounds upon heating may undergo degradation and oxidation. The decrease at these optimal conditions of the total phenol, flavonoid contents, and antioxidant activities (FRAP) of the extracts could also suggest that some phenolics were rendered less extractable, due to the interaction of the phenolic compound with other food components, such as proteins [34]. Mwangwela and Waniska [13] reported that infrared heating of cowpea resulted in crosslinking of the cytoplasmic matrix, and these crosslinks formed may have involved the interaction of cowpea phenolic compounds with macromolecules such as proteins. The polymerization and/or complexation of leguminous seeds’ phenolic compounds with its macromolecules such as proteins have been reported [30,31,35] and could have reduced its availability and extractability.

### 3.6. Level of Individual Phenolic Compounds

In Table 6, eleven phenolic compounds in the methanol extracts from the control samples and cowpea processed at optimized conditions were identified by analyzing the MS^2^ fragment information and UV spectral characteristics of those compounds and then comparing them with standards and previously reported results. Similarly, Kayitesi [7] reported that phenolic acids including protocatechuic acid, *p*-coumaric acid, and ferulic acid and flavonoids, such as catechin, rutin, and quercetin are the most abundant in cowpeas. The result also shows that phenolic compounds, such as luteolin, ferulic acid, gallic acid, and vanillic acid significantly increased in the extract of treated cowpea at optimized condition by 42, 78, 48, and 49%, respectively.

Compounds such as taxifolin, sinapic acid, and caffeic acid showed a significant (*p* < 0.05) reduction in the extract of treated cowpea at optimized conditions by 50, 71, and 48%, respectively. Others, such as kaempferol, quercetin acid, and apigenin identified did not show any statistically significant (*p* ≥ 0.05) increase or reduction in the extract of treated cowpea at optimized conditions. A similar trend of increase, decrease, or no change were observed in phenolic compounds of infrared heated cowpea alone or followed by the cooking of treated seeds [7]. The author attributed the increase to the release of phenolic-linked cell wall components, and the decrease could be due to the crosslinking of phenolic compounds with macromolecules to form insoluble complexes, causing a reduction in their extractability.

### 3.7. Phytate and Oxalates

The phytate (9.99 mg/g) and oxalates (1.73 mg/g) of optimized infrared heated cowpea was significantly lower than the control by 24% and 42%, respectively (Figure 3). The substantial reduction in the optimized infrared heated cowpea sample could be associated with thermal inactivation of the antinutrients.

It has been previously reported that infrared heating causes a reduction in antinutrients, such as phytic acids, tannin, trypsin inhibitor, and oligosaccharides levels of legume seeds due to rapid internal heating as a result of increased molecular vibrations and intermolecular friction within the seeds [14,15,36]. Xu and Chen [37] reported that trypsin inhibitors are unstable and incorporated into protein aggregates at cooking temperatures compared to the Bowman–Birk trypsin inhibitors, which are more heat stable.

## 4. Conclusions

In conclusion, the optimization of infrared processing conditions for cowpea seeds using response surface methodology showed that the optimum moisture level, heating temperature, and time were 45%, 185 °C, and 5 min, respectively. Phenolic compounds, such as luteolin, ferulic acid, gallic acid, and vanillic acid were the most abundant, and although some phenolic compounds decreased with infrared heating in the optimal samples, the optimized infrared cowpea sample still exhibited antioxidant properties capable of scavenging free radicals. The cowpea grains produced at these optimized conditions displayed improved physical properties (bulk density and water absorption), pectin solubility, and reduced antinutrient activity (oxalates and phytate) as compared to the untreated (control). Cowpea grains made from the optimized conditions could be nutritious and an important intermediate raw material in the production of infant products.

## Figures and Tables

**Figure 1 molecules-26-06137-f001:**
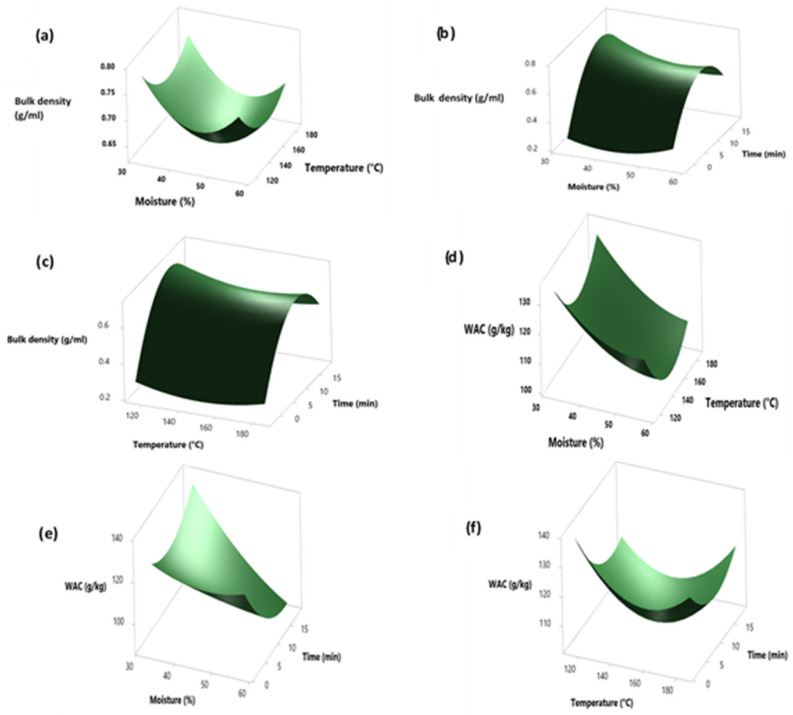
Surface plots of bulk density and water absorption capacity (WAC). Plots: (**a**): effect of moisture–temperature, (**b**): effect of moisture–time, (**c**): effect of temperature–time on bulk density; (**d**): effect of moisture–temperature, (**e**): effect of moisture–time, (**f**): effect of temperature–time on WAC.

**Figure 2 molecules-26-06137-f002:**
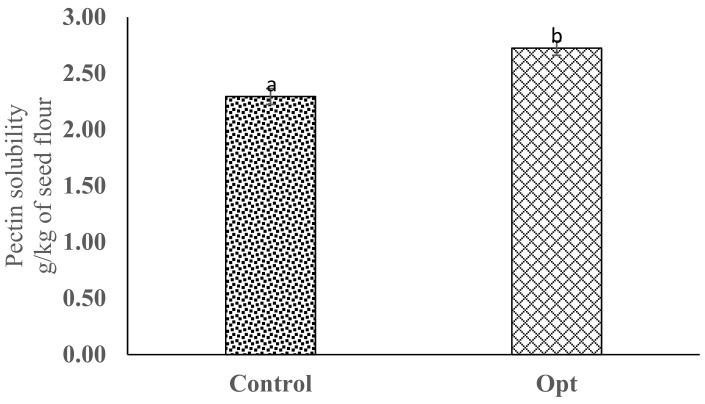
Pectin solubility content of cowpea processed at optimized conditions. Error bars indicate standard deviation (N = 3). Bars with different alphabets indicate significant difference (*p* ≤ 0.05).

**Figure 3 molecules-26-06137-f003:**
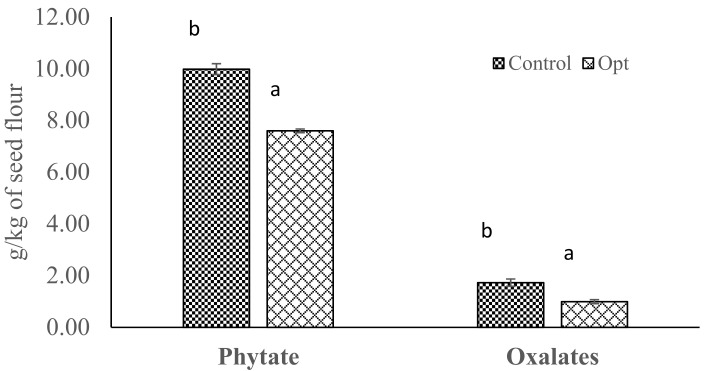
Phytate and oxalates content of cowpeas processed at optimized conditions. Error bars indicate standard deviation (N = 3). Bars with different alphabets indicate significant difference (*p* ≤ 0.05).

**Table 1 molecules-26-06137-t001:** Experimental conditions for optimization of moisture and infrared heating conditions (infrared heating temperature and time).

Levels						
Factors	Codes	−α	−1	0	1	+α
Moisture level (%)	X_1_	32.6	40	45	54	57.3
Infrared temperature (°C)	X_2_	114.7	130	150	170	185.3
Infrared time (min)	X_3_	NA *	2	8	14	18.5

* NA—not applicable. The design of experiments based on the CCD yielded a negative value (−2.5 min) for—α time; hence, experimental conditions containing these values were excluded from the experiments.

**Table 2 molecules-26-06137-t002:** Bulk density and water absorption capacity (WAC) of moisture conditioned follow by infrared heating of cowpea seeds for different experimental runs.

Experimental Runs	Moisture	Temperature	Time	Bulk Density (g/mL)	WAC (g/kg)
1	40.00	130.00	2.00	0.67	125.54
2	40.00	130.00	14.00	0.64	114.66
3	40.00	170.00	2.00	0.66	126.83
4	40.00	170.00	14.00	0.62	126.78
5	54.00	130.00	2.00	0.62	126.78
6	54.00	130.00	14.00	0.63	99.96
7	54.00	170.00	2.00	0.62	125.85
8	54.00	170.00	14.00	0.60	102.39
9 (C)	45.00	150.00	8.00	0.62	103.38
10	32.65	150.00	8.00	0.67	118.06
11	57.35	150.00	8.00	0.61	95.45
12	45.00	114.72	8.00	0.64	125.62
13	45.00	185.28	8.00	0.60	103.84
15	45.00	150.00	18.58	0.62	98.55
16 (C)	45.00	150.00	8.00	0.62	108.60

C: replicated center-point to improve the precision of the model.

**Table 3 molecules-26-06137-t003:** Coefficient of regression for the different mathematical models obtained.

Coefficient	Bulk Density	WAC
*β* _0_	0.74	1248.00
*β* _1_	−0.00	−18.88
*β* _2_	0.00	−9.08
*β* _3_	−0.01	−4.17
*β* _11_	0.00	0.17
*β* _22_	−0.00	0.03
*β* _33_	0.00	0.36
*β* _12_	−0.00	0.02
*β* _13_	0.00	−0.02
*β* _23_	−0.004	−0.02
*R*^2^ (%)	95.08%	92.15%
*R*^2^*adj* (%)	93.56%	89.71%
Residual	0.01	4.09

*β* represents the coefficients of equations of the different models with *β*_0_ representing the constant term; *β*_1_, *β*_2_, and *β*_3_ represent the linear effects of moisture level, infrared heat treatment temperature, and time, respectively; *β*_11_, *β*_22_, and *β*_33_ represent their quadratic effects; and *β*_12_, *β*_13_, and *β*_23_ represent the interactions between bulk density and water absorption capacity.

**Table 4 molecules-26-06137-t004:** CCD (central composite design) regression model fit factor effects for bulk density and water absorption capacity of infrared heated cowpea seeds.

	Bulk Density		WAC	
Term	Effect	*p*-Values *	Effect	*p*-Values *
Linear effects (L)				
Constant	-	0.000	-	0.000
Moisture	−0.029	0.000	−2.258	0.186
Temperature	−0.018	0.000	−6.881	0.000
Time	−0.019	0.000	−24.249	0.000
Quadratic effects (Q)				
Moisture × Moisture	0.006	0.329	16.650	0.001
Temperature × Temperature	−0.006	0.347	21.660	0.000
Time × Time	0.006	0.404	25.710	0.000
Interactive effects (I)				
Moisture × Temperature	−0.002	0.368	5.466	0.002
Moisture × Time	0.014	0.000	−1.871	0.269
Temperature × Time	−0.009	0.001	−4.026	0.022

* Statistically significant (*p* ≤ 0.05) factors. L—linear effect of moisture, temperature, and time, Q—quadratic effect of moisture, temperature, and time, I—interaction effect moisture and temperature, moisture and time, and temperature and time.

**Table 5 molecules-26-06137-t005:** Total phenolic content, total flavonoid content, and antioxidant property of extracts of cowpea processed at optimized conditions.

Samples		
	Control	Treated (Opt)
TPC (mg CE/g)	1.10 ^b^ (0.02)	0.47 ^a^ (0.04)
TFC (mg CE/g)	4.23 ^b^ (0.08) ^1^	1.66 ^a^ (0.07)
Antioxidant properties		
FRAP (mg GAE/g)	8.21 ^b^ (0.44)	2.32 ^a^ (0.01)

^1^ Means and standard deviation *n* = 3. Values with different superscript alphabets in rows are significantly different (*p* < 0.05). Sample extracts: Control (0 min), Opt: Cowpea produced at optimum condition; moisture level, infrared heating temperature and time of 45%, 185 °C, and 5 min, respectively. TPC, total phenolic content; TFC, total flavonoid content.

**Table 6 molecules-26-06137-t006:** Individual free phenolic compounds (µg/mL) identified in the extracts of cowpea processed at optimized conditions.

No	Phenolic Compounds	Control Sample (µg/g)	Treated (opt) (µg/g)
1	Kaempferol	6.60 ^a^ (0.09)	2.80 ^a^ (0.31) ^1^
2	Luteolin	6.52 ^a^ (0.01)	11.32 ^b^ (0.08)
3	Ferulic acid	5.08 ^a^ (0.54)	22.96 ^b^ (1.23)
4	Taxifolin	979.12 ^b^ (18.67)	494.24 ^a^ (1.96)
5	Apigenin	2.16 ^a^ (0.13)	3.80 ^a^ (0.49)
6	Quercetin	1.20 ^a^ (0.11)	0.68 ^a^ (0.03)
7	p-Coumaric acid	83.84 ^a^ (4.95)	52.00 ^a^ (1.46)
8	Sinapic acid	92.96 ^b^ (0.52)	26.72 ^a^ (0.88)
9	Caffeic acid	289.76 ^b^ (0.42)	130.80 ^a^ (1.04)
10	Gallic acid	169.64 ^a^ (0.23)	328.48 ^b^ (1.75)
11	Vanillic acid	349.68 ^a^ (1.04)	684.44 ^a^ (4.18)

^1^ Means and standard deviation *n* = 3. Values with different superscript alphabets in rows are significantly different (*p* < 0.05) for the levels of individual phenolic compounds among the control and Opt. Sample extracts: Control (0 min), Opt: Cowpea produced at optimum condition; moisture level, infrared heating temperature and time of 45%, 185 °C, and 5 min, respectively.

## Data Availability

Not available.

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
