# Peer review of "Optimization of Infrared Heating Conditions for Precooked Cowpea Production Using Response Surface Methodology"

_molecules, 2021, doi:10.3390/molecules26206137_

Round 1

Reviewer 1 Report

It is quite fair study, however it needs some improvement. The basic thing is to define control sample in materials and methods

Line 11 - delate the parentheses
Line 21-22 - what kind of Inhibitors of phytates and oxalates authors mean? Rephrase this sentence because it's not true
Unify: tabletop or table-top 
Table 6 -  correct the first four  lines in order to look like the rest
Capital letter in caption (figure 3) in oxalates is not necessary

Author Response

Comments and Suggestions for Authors

Comment: It is quite fair study; however, it needs some improvement. The basic thing is to define control sample in materials and methods.

Response: We thank the reviewer and as suggested the control has now been defined in the material and method.

Comment: Line 11 - delate the parentheses

Response: parentheses in “using” was not original on the document submitted 

Comment: Line 21-22 - what kind of Inhibitors of phytates and oxalates authors mean? Rephrase this sentence because it's not true

Response: “the inhibitors” now changed to “these antinutrients”

Comment: Unify: tabletop or table-top

Response: “tabletop” now revised through the MS as “table-top” 

Comment: Table 6 - correct the first four lines in order to look like the rest

Response: Table 6, fourth line and the whole table now revised as suggested.

Comment: Capital letter in caption (figure 3) in oxalates is not necessary

Response: Oxalates now revised as “oxalates”

Reviewer 2 Report

Minor remarks

  • The references should be prepared according to the Instruction for Authors.
  • Provide a blank space between quantity and unit, only in the case of percentage.
  • In Table 3, the numbers should be rounded in the same decimal places.
  • Some references in the manuscript are not numbered.
  • All other minor remarks are given in the manuscript.

Major remarks

  • Each reference should be discussed separately so that lumping references is not necessary.
  • A literature review is commonly composed of older references, so I recommend providing newer ones.
  • The novelty of the manuscript should be better defined.
  • The range of infrared time is not adequate because it is not real. You should define a new real range of this parameter.
  • Equation 1 should be retyped. Please, insert the terms referring to the third analyzed parameter (C). Also, the given level (-alpha) is not depicted later in table 2. Please, check this. Do you have any explanation?
  • The normal probability plot and Cook’s distance for the proposed model should be presented. The way how to do that is well present in the following papers: DOI: 10.3390/antiox8080248; DOI: 3390/biom11020225. These papers can be included in the reference list.
  • It is well to know, that total phenolic content such as presented in this manuscript does not represent the real content of phenolic compounds. It is true that phenolic compounds react with Folin–Ciocalteu reagent, but they are not the only ones. Also, vitamin C, some amino acids, peptides, reducing sugars, organic acids, Maillard reaction products, and some more compounds can react with Folin–Ciocalteu reagent. That is well presented in the following paper DOI: 10.1007%2Fs11081-020-09565-0. This paper can be included in the reference list. Having in mind this fact, it is desirable to write the content of antioxidants instead of total phenolic content.

Author Response

Reviewer 2

Minor remarks

Comment: The references should be prepared according to the Instruction for Authors.

Response: References have now been revised according to the Instruction for Authors.

Comment: Provide a blank space between quantity and unit, only in the case of percentage.

Response: blank space between quantity and unit has been revised throughout the MS, only in the case of percentage as suggested.

Comment: In Table 3, the numbers should be rounded in the same decimal places.

Response: Table 3 has now been revised as suggested

Comment: Some references in the manuscript are not numbered.

Response: The references has now been revised as suggested throughout the MS.

Comment: All other minor remarks are given in the manuscript.

Response: All the minor remarks has now been revised through the entire MS as suggested.

Major remarks

Comment: Each reference should be discussed separately so that lumping references is not necessary.

Response: To the best of our knowledge the manuscript has been revised so that lumping references is avoided.

Comment: A literature review is commonly composed of older references, so I recommend providing newer ones.

Response: Reference has been revised and older ones replaced with newer ones where applicable to the best of our knowledge older ones remain are the only available source of information. 

Comment: The novelty of the manuscript should be better defined.

Response: The novelty has now clearly been defined and highlighted in green at the concluding part of the introduction.

Comment: The range of infrared time is not adequate because it is not real. You should define a new real range of this parameter.

Comment: Equation 1 should be retyped. Please, insert the terms referring to the third analyzed parameter (C). Also, the given level (-alpha) is not depicted later in table 2. Please, check this. Do you have any explanation?

Comment: The normal probability plot and Cook’s distance for the proposed model should be presented. The way how to do that is well present in the following papers: DOI: 10.3390/antiox8080248; DOI: 3390/biom11020225. These papers can be included in the reference list.

Response: We thank the reviewer for this suggestion, using Minitab we have plotted the Normal residual plots (Supplementary data), discussed it in our result and using DOI: 3390/biom11020225 as suggested by the reviewer. However, because we different software (MINITAB) to the reference provided by the reviewer for our plots, we have been unable to generate the Cook Plots.

Comment: It is well to know, that total phenolic content such as presented in this manuscript does not represent the real content of phenolic compounds. It is true that phenolic compounds react with Folin–Ciocalteu reagent, but they are not the only ones. Also, vitamin C, some amino acids, peptides, reducing sugars, organic acids, Maillard reaction products, and some more compounds can react with Folin–Ciocalteu reagent. That is well presented in the following paper DOI: 10.1007%2Fs11081-020-09565-0. This paper can be included in the reference list. Having in mind this fact, it is desirable to write the content of antioxidants instead of total phenolic content.

Response: We totally agree with the reviewer and appreciate the highlight provided. We have revised the statement as suggested and cited the reference DOI: 10.1007/s11081-020-09565-0 as a scientific rationale to support our result and findings.

Round 2

Reviewer 2 Report

The manuscript can be accepted in the present form.